Elephant seal dive behaviour responds consistently to changes in foraging success regardless of sex or ocean habitat

Green David B. davo.b.green@gmail.com 1 2
Bestley Sophie 1 2 3
McMahon Clive R. 1 2 4 5
Lea Mary-Anne 1 2
Harcourt Robert G. 5
Guinet Christophe 6
Hindell Mark A. 1 2
1 Australian Centre for Excellence in Antarctic Science (ACEAS), University of Tasmania , Hobart , TAS , Australia
2 Institute for Marine and Antarctic Studies, University of Tasmania , Hobart , TAS , Australia
3 Australian Antarctic Program Partnership, University of Tasmania , Hobart , TAS , Australia
4 IMOS Animal Tagging, Sydney Institute of Marine Science , Mosman , NSW , Australia
5 School of Natural Sciences, Macquarie University , North Ryde , NSW , Australia
6 Centre d’Etudes Biologiques de Chizé, CNRS , Villiers en Bois , France
Bernardino Angelo
Electronic publication date: 2025 Dec 17
Publication date: 2025
Volume: 13
Electronic Location ID: e20378
Received 2025 Jul 2; Accepted 2025 Oct 20
Copyright: ©2025 Green et al.
Copyright year: 2025
Copyright holder: Green et al.
License: This is an open access article distributed under the terms of the Creative Commons Attribution License, which permits unrestricted use, distribution, reproduction and adaptation in any medium and for any purpose provided that it is properly attributed. For attribution, the original author(s), title, publication source (PeerJ) and either DOI or URL of the article must be cited.
License URL: https://creativecommons.org/licenses/by/4.0/

Keywords: Southern elephant seal, Southern Ocean, Predator, Prey, Drift rate, Foraging behaviour, Dive effort, Body condition

Funding: Australian Research Council Special Research Initiative, Australian Centre for Excellence in Antarctic Science SR200100008 Australia’s Integrated Marine Observing System (IMOS) NIWA New Zealand and Australian Antarctic science grants This research was supported by the Australian Research Council Special Research Initiative, Australian Centre for Excellence in Antarctic Science (Project Number SR200100008). Funding for the fieldwork was supplied by Australia’s Integrated Marine Observing System (IMOS), and NIWA New Zealand and Australian Antarctic science grants. Data for this study were sourced through the Australian Integrated Marine Observing System (IMOS). IMOS is enabled by the National Collaborative Research Infrastructure Strategy (NCRIS). The funders had no role in study design, data collection and analysis, decision to publish, or preparation of the manuscript.

==============================
Understanding how air-breathing diving animals moderate their dive behaviour when foraging successfully is foundational in the study of their foraging ecology. Yet, this fundamental relationship remains unresolved with previous research pointing to inconsistent relationships, differing nominally according to sex, habitat type and scale. Empirically testing the relationships between dive effort responses and foraging success is further hampered because of challenges obtaining concurrent measures of behavioural responses and foraging success at sea. We compiled a multi-decadal dive dataset from 609 southern elephant seals, including their dive responses (transit rate, and relative dive and surface recovery duration) and buoyancy—changes in which provide an indirect measure of body condition change and foraging success. Using this dataset, we tested how seal dive behaviour alters when foraging remotely at sea. We found that as foraging success increased, seals increased transit (ascent, descent) rates and decreased relative dive durations for a given depth, with no response in surface recovery. Our results were consistent across sexes and foraging habitats, and account for the general effects of buoyancy on dive behaviour. The homogeneity of these findings suggests that there is a general functional response in which elephant seals perform, on average, shorter, steeper dives during periods of successful foraging. Importantly, we can align these results with predictions from the marginal value theorem (MVT), that a forager should remain in a patch only until gains drop below the neighbourhood mean. Our findings have broad-based implications for how ecologists interpret dive responses of wild marine animals, demonstrating the value of seeking independent in situ information on foraging success.

Introduction

How animals respond to the distribution of food resources is foundational to the study of foraging ecology (Stephens & Krebs, 1986) and the way we measure and quantify animal performance (Beltran et al., 2023). In marine systems, a long-standing goal has been to understand how air-breathing diving marine animals modify their behaviour to forage successfully while under the constraints of net oxygen depletion (Kooyman, 1989).

Across breath-hold diving animals, the cycle of any given dive includes three distinct phases: a bottom phase associated with active foraging; a transit phase, bookending the bottom phase, during which an individual commutes between the surface and depth; and finally a post-dive surface recovery phase when an animal reoxygenates. This structure makes it convenient to consider these animals as central-place foragers, commuting between the surface (central place) and depths (foraging areas). Measurements from these phases (dive metrics, e.g., descent and ascent rates, bottom and surface recovery times) provide a useful estimate of animal foraging effort (Bestley et al., 2015; Green et al., 2020), which itself might be used as an indirect measure of habitat quality in the absence of direct observations of prey capture or body condition change (Heerah et al., 2014; Heerah et al., 2015). However, despite the potential utility of using effort estimates in lieu of direct foraging success measures, these efforts are limited by the fundamental uncertainty of whether foraging effort should increase or decrease during periods of successful foraging.

Diving from a central place (surface), animals are generally predicted to maximise time in the bottom phase when food is encountered (Houston & Carbone, 1992), whilst simultaneously minimising time in transit (without increasing speed-related drag excessively; Biuw et al., 2003) and at the surface. For the latter, it is reasonable to assume that actively foraging divers would reduce ‘wasted’ time spent in transit or at the surface (ignoring possible increases in energetic costs, e.g., through faster swimming), given these phases are not generally associated with prey captures. Under these conditions, we can predict that when a diver is foraging successfully, it would reduce time between dives and increase transit rates between the surface and depths to maximise prey encounter opportunities. However, for these animals to then reallocate all of that ‘saved’ time to the bottom phase only makes energetic sense if this also leads to a monotonic increase in gains (Austin et al., 2006; Robinson et al., 2007; although see also Thompson & Fedak, 2001). This might be the case in relatively uniformly distributed prey fields but is unlikely to hold true in environments where prey are more patchily distributed than this would allow.

Pelagic marine environments are highly patchy, and over any two consecutive dives an animal may encounter markedly different prey densities. Confronted with very low prey densities, it is reasonable to assume that an animal would terminate a dive early. However, even when encountering dense prey patches, theory predicts that attempting to maximise the bottom phase of every dive is unlikely to be the most efficient energetic strategy. Dive-related interpretations of the marginal value theorem (MVT) hold that an animal should terminate a dive early if prey encounter rates are or drop below the average for a given “neighbourhood” of dives (Charnov, 1976). This means that even in very profitable environments where each consecutive dive offers high prey encounters (high neighbourhood quality), average dive bottom time may be shorter because prey encounters are quickly depleted to that of the neighbourhood mean. In contrast, where dives with high prey encounter rates are interspersed amongst dives of generally low prey encounter rates (low neighbourhood quality), it would take longer for prey encounter rates to be depleted to this lower “neighbourhood” mean, and we subsequently would expect higher average bottom durations.

Identifying whether there are consistent patterns of dive effort responses under changing levels of foraging success is critical if we are to develop reliable measures of habitat quality. Yet, empirical evidence so far remains equivocal, with contrasting patterns often linked to inter- or intra- sex foraging strategy (Beck et al., 2003) or habitat use (Jessopp, Cronin & Hart, 2013) and/or scale (e.g., Austin et al., 2006; Thums et al., 2013). As an example, benthic and pelagic foragers might schedule their bottom duration differently when encountering prey fields of variable spatial structuring spanning from uniformly distributed, to highly patchy. Similarly, increasing foraging success is associated with both longer and shorter bottom phase durations, depending on whether these metrics are measured at the scale of dives or integrated across dives (Foo et al., 2016; Watanabe, Ito & Takahashi, 2014). So, while we could comfortably predict that divers should uniformly attempt to minimise transit times and surface recovery when foraging successfully, we might expect the bottom time response to be more context specific.

Southern elephant seals (Mirounga leonina) are one of the few species for which a measure of the dive cycle and an estimate of foraging success can be obtained in situ. Approximately 3–4% of daily dives contain segments when seals are inactive and drift passively in the water column (Arce et al., 2019; Biuw et al., 2003; Crocker, Boeuf & Costa, 1997; Thums, Bradshaw & Hindell, 2008). During these segments, a seal’s buoyancy determines the rate and direction of its vertical displacement through the water column. Buoyancy itself is primarily a function of a seal’s body condition (blubber:lean tissue ratio), with thinner seals descending and fatter seals ascending when drifting. This link between buoyancy and body condition means that incremental changes in buoyancy can be used as a measure of net energy gain (Crocker, Boeuf & Costa, 1997), and subsequently foraging success.

Complicating matters for diving marine animals, like elephant seals, is the role that buoyancy plays in influencing an individual’s physiology and subsequent dive characteristics (e.g., Jouma’a et al., 2016; Piot et al., 2023). As elephant seals improve body condition and approach neutral buoyancy, their dives become more energetically efficient (e.g., Hassrick et al., 2010; Piot et al., 2023; Richard et al., 2014), reducing oxygen consumption. With lower oxygen costs, seals can perform longer dives. This means we could expect buoyancy to exert a first order control on dive behaviour, extending to the descent and ascent phases also (Beck, Bowen & Iverson, 2000; Richard et al., 2014; Webb et al., 1998). However, we expect variations in dive behaviour extending beyond that accounted for by buoyancy, to be a response to variable prey conditions encountered by foraging seals.

Here we tested for consistent relationships between a suite of dive effort metrics and foraging success, using a dataset from over 600 tracked southern elephant seals. Elephant seals display strong sexual dimorphism, leading to males and females having very different life history strategies and foraging ecologies (e.g., McIntyre et al., 2010). Additionally, the species is extremely wide ranging, dispersing widely across the Southern Ocean to both deep oceanic environments as well as shallower shelf waters (Hindell et al., 2021). This means that within any population there exist numerous foraging strategies (Abrahms et al., 2018), that vary by sex and individual use of oceanic (pelagic) and shelf (benthic) habitats (Hindell et al., 2021). These different habitats likely show varied spatial structuring spanning from relatively uniform (shelf) to highly patchily distributed prey fields (oceanic) (e.g., Riverón et al., 2021). Contextualising these relationships against the backdrop of optimal foraging theory, we aimed to understand the nature of the dive effort response to foraging success across sexes and habitats, and whether we could identify consistent responses between dive effort metrics and foraging success, regardless of sex and foraging habitat.

Materials and Methods

Tagging dataset

We compiled 20 years of at-sea tracking data (2004–2023) from 265 female and 344 male southern elephant seals fitted with a CTD-SRDL-9000 (Conductivity-Temperature-Depth Satellite Relay Data Logger–Sea Mammal Research Unit, St Andrews, UK) across five sites: Iles Kerguelen (49.35°S, 70.22°E; n = 472), Macquarie Island (54.50°S, 158.95°E; n = 39), Campbell Island (52.55°S, 169.09°E; n = 16), Davis Station (68.58°S, 77.87°E; n = 57) and Casey Station (66.28°S, 110.53°E; n = 25). The large number of tracks represented here were collected primarily as part of global ocean observing initiatives, delivering critical biophysical ocean observations into long-term programs including MEOP (Marine Mammals Exploring the Oceans Pole to Pole) and the GOOS (Global Ocean Observing System), both aimed at monitoring global ocean physics and the state of marine ecosystems (McMahon et al., 2021; Treasure et al., 2017). The significant size of this dataset has concurrently been used to investigate the circumpolar foraging distribution and behaviour of elephant seals, providing unique insights into the species’ biology beyond the capacity of smaller-scale tracking projects (e.g., Green et al., 2020; Hindell et al., 2016; Hindell et al., 2022; Hindell et al., 2021; McMahon et al., 2025).

Tag deployment procedures

All tag deployment procedures were conducted using methods recognised as best practice for the anaesthesia and instrumentation of seals (McMahon et al., 2025). Prior to deployment, seals were restrained using a head bag and subsequently chemically sedated with a mixture of tiletamine and zolazepam (0.5 mg/kg) administered intra-venously following methods detailed in McMahon et al. (2000). Thereafter the tag was glued to the pelage on the seal’s head (Field et al., 2012). The seal was then released and monitored from a distance until it had regained full mobility. The CTD-SRDLs remained on the seals until they either fell off at sea or during the annual moult. All tagging procedures were approved and executed under University of Tasmania Animal Ethics Committee guidelines (A0028182), the Comité d’éthique Anses/ENVA/UPEC (no. APAFiS: 21375), and the Australian Antarctic Animal Ethics Committee (grant nos. AAS 2265 and AAS 2794). Fieldwork was conducted under permit approval of, and logistically supported by the respective national program administering each field site, namely the French Polar Institute Paul-Émile Victor (Iles Kerguelen; IPEV 109 & 1201), the Tasmanian Government (TFA 22459; Macquarie Island), the Australian Antarctic Division (AAS 2265 & AAS 2794; Davis Station and Casey Station) and the New Zealand Department of Conservation (SO-32768-MAR; Campbell Island). We removed from our subsequent analyses all tracks with <5 days of data (n = 18), with 558 (94%) of the remaining 591 seals having tags that transmitted for more than 30 days.

While at sea, CTD-SRDLs provided 2–15 ARGOS satellite location estimates per day and a random sample of 1 to 186 dive profiles per day (Boehme et al., 2009; Fedak, Lovell & Grant, 2001). Depth was sampled every 4 secs, but individual dives were summarized onboard the device providing five time-depth segments, separated using a broken-stick algorithm which identified the four inflection points that best represent dive profile shape (Photopoulou et al., 2015). Along with these summarised dive profiles, CTD-SRDLs also relayed measurements of maximum dive depth, dive duration, and post-dive surface interval.

Track path estimation

To estimate the most likely path and dive locations for individual seals, accounting for the errors associated with ARGOS satellite location estimates, we used a continuous time state-space model assuming a maximum horizontal speed of 4 m ⋅ s−1, implemented using the aniMotum package (Jonsen et al., 2023). We fitted the model using a correlated random walk process, which we identified as the most appropriate based on initial assessment of computed one-step-ahead residuals (Jonsen et al., 2023). The state-space model provides estimates of locations, with associated uncertainty, at user-specified times. Using this method, fitted position estimates were obtained at the observation times of the ARGOS fixes, and predicted position estimates were obtained at the timestamp of each summarised dive. All data processing and statistical analyses were conducted in R version 4.2.0 (R Core Team, 2023).

Estimating foraging success from changes in drift rate

We estimated drift rates (buoyancy) for each seal using the slimmingDive package (Arce et al., 2019; Arce et al., 2022). The method only considers dives deeper than 100 m, accounting for the effects of residual air in the lungs at shallower depths (Arce et al., 2019). Thereafter, slimmingDive uses a Kalman filter to evaluate the probability (Z) that a drift rate observation is inside or outside the most likely trajectory of the drift rate time series (see Arce et al., 2019 for a full description of the method). Vertical speeds were obtained from all drift segments for which Z > 0.5.

Assimilation of prey into blubber, and its subsequent effects on buoyancy may take up to about 20 h before being reflected as changes in drift rate (Biuw et al., 2003; Krockenberger & Bryden, 1994), indicating that daily drift rate estimates are the finest resolution that can be usefully calculated. However, changes in drift rate have also previously been shown to be visible only after 5–7 days or longer (Biuw et al., 2003; Thums, Bradshaw & Hindell, 2008). We therefore computed an unweighted mean drift rate for time windows spanning 1, 2, 3, 7 and 10 days. We evaluated foraging success through estimates of drift rate change (ΔDR), i.e., by taking the difference between current drift rate and that from the previous time step (ΔDR = drift rate[n] − drift rate[n−1]), calculated for each of these temporal windows. We consider foraging success measured in this way to be analogous to the mean net energy gain across neighbourhoods of dives, integrating information on energy quality, density and availability of prey encountered (albeit offset by foraging costs) over a given time interval (Crocker, Boeuf & Costa, 1997; Biuw et al., 2003; Yong et al., 2024).

Computing dive metrics as indicators of behaviour

We considered only dives with a maximum depth >15 m and a duration >5 min (Allegue et al., 2023). We also omitted any dives containing: (1) missing depth values, or multiple depth values for a given time, (2) any dive segments with vertical displacements >4 m ⋅ s−1, (3) a total duration longer than 90 min, (4) a maximum depth >2,000 m, or (5) a surface time exceeding 10 min. These thresholds effectively removed outlier dives and were chosen based on inspections of histograms for each variable. These values are slightly more conservative than those previously suggested by Cox et al. (2018).

Using this dataset, we computed four dive metrics representing three distinct dive stages, namely: descent and ascent rate (transit between surface and bottom phase and vice versa), dive residual (relative dive duration given the maximum depth—see below), and surface residual (relative post-dive recovery duration given dive duration). To compute descent and ascent rates, we respectively calculated the absolute vertical rates of change between the surface and first dive inflection point, and between the last dive inflection point and the surface.

Dive duration is strongly linked to the maximum attained depth, given that the descent and ascent phases of each dive must naturally increase with dive depth. Dive duration should therefore be considered together with depth to determine relative dive effort (Bestley et al., 2015). We did this by calculating the dive residual, being the residual of the regression between dive depth and dive duration (Bestley et al., 2015); a practical measure of whether a dive is relatively long or short given its depth. Similarly, the post-dive surface recovery interval is expected to increase with increasing dive duration. The surface residual represents the difference between the observed surface recovery interval and the predicted minimum surface recovery interval for a dive of that duration (Bestley et al., 2015).

These dive metrics could then be compared against our independent estimates of foraging success. As with foraging success, we computed an unweighted running mean of each metric across time windows spanning 1, 2, 3, 7 and 10 days. This allowed us to investigate the relationships between dive response metrics and foraging success across a range of temporal scales.

Modelling dive behaviour as functions of foraging success and buoyancy

We used generalised additive mixed models (GAMMs; Wood, 2017) to estimate the relationship between our dive behaviour metrics (descent and ascent rates, and dive and surface residuals) and foraging success (drift rate change, ΔDR). These models account for multiple measurements on a single sampling unit (i.e., many observations per seal through time) and allow the within-individual errors to be correlated and/or have unequal variances. These are modelled as random effects.

We expected buoyancy to exert a first order control on dive behaviour, extending to the descent and ascent phases also (Beck, Bowen & Iverson, 2000; Richard et al., 2014; Webb et al., 1998). However, after accounting for buoyancy (Drift Rate (DR)) control on dive behaviour, we expect further variation in dive behaviour to be influenced by energy gain (ΔDR) over a given time period, our proxy for foraging success. Noting this, successful modelling of the relationship between dive behaviour metrics and foraging success requires simultaneous correction for the confounding effects of buoyancy.

Given the different foraging success rates between shelf and oceanic foraging seals (Arce et al., 2022), we considered these two habitats independently. The 1,500 m isobath is generally considered to represent the shelf break (e.g., Malpress et al., 2017). We therefore classified the average location for a given temporal window as being respectively either shelf or oceanic habitat depending on whether waters were shallower or deeper than this isobath (Fig. 1).

Figure 1 Individual seal tracks (females: n = 265; males: n = 344) coloured by sex, where green denotes males and orange denotes females.

Lighter shades of green and orange show oceanic habitats while dark shades represent shelf habitats. The light grey contour gives the location of the 1,500 m isobath and represents the threshold between oceanic and shelf habitats. Bathymetry data were obtained from the GEBCO 2021 bathymetry product, and continent boundaries were obtained from the R package “rnaturalearth” (South, Michael & Massicotte, 2017).

We fitted separate models for our measures of sex, foraging habitat, and temporal window. For each model, we fitted our dive behaviour metric as the behaviour with two predictors namely ΔDR and DR. To aid comparison of coefficients across sex and habitat, ΔDR was fitted as a linear predictor. We included both a random slope and intercept on seal ID, with the random slope allowing the ΔDR relationship to vary seal-to-seal. To account for serial autocorrelation the errors were specified to follow a first-order autoregressive process within seal (AR1 correlation structure, corCAR1, on a continuous time covariate, days since departure). Drift rate was fitted using a thin-plate spline smooth, to account for the component of variability in our four dive response metrics that is driven by buoyancy (as described above) and allowing the shape of the emergent relationships to be flexible (non-prescribed). Please refer to ESM Table S1 for a detailed list of all modelled combinations of dive metric, sex, habitat and temporal window.

While these models may not fully decouple measures of foraging success and dive metrics, both calculated from dive profiles, we contend that they are fit for purpose and represent the best achievable estimate of both variables when dealing with heavily summarised satellite-relayed data. We also note that we could have combined individual models for sex and foraging habitat by specifying these as interaction terms on ΔDR and DR. However, preliminary attempts at doing so increased model complexity, reduced interpretability and led to convergence issues. Nonetheless, outputs from preliminary models that did converge indicated that the added model complexity would not change our findings. All GAMMs were fitted using the mgcv package (Wood, 2017). Model performance was evaluated based on reported goodness of fit and diagnostic plots produced through the gam.check function (see ESM for model summaries and diagnostic plots).

Results

Spatial and temporal variance of seal buoyancy and dive response metrics

Overall, seals departing deployment locations were negatively buoyant, and showed a progressive increase in buoyancy (body condition) over the course of their post-moult migration, with some individuals approaching neutral buoyancy by approximately the third month at sea (Fig. 2). Concurrent with increasing buoyancy, seals also displayed progressive changes in dive characteristics. Descent and ascent rates decreased noticeably over the first three months as seals gained buoyancy, by contrast dive residuals increased. Surface residuals showed a weak tendency to increase over time.

Figure 2 Time series showing changes in mean daily drift rate and mean daily dive effort metrics (descent rate, dive residual, ascent rate and surface residual) across days since departure.

Values for all females are shown by orange points and males as green points. Time series of one individual tracked seal is illustrated by the black line and points. When approaching neutral buoyancy (drift rates of ∼0 m⋅s−1), seal buoyancy tends to show rapid switching between negative and positive values (Biuw et al., 2003), this accounts for the relatively few values recorded within the −0.1–0 m⋅s−1 range for the top panel. Y-axis scales have been crimped to the scale of dominant patterns.

Relationships between dive behaviour, foraging success and drift rate

Accounting for the nonlinear relationship between all dive behaviour metrics and drift rate (Fig. 3) we found that all metrics, excluding surface residual, were linearly correlated with foraging success (i.e., ΔDR, the change in drift rate over time). Across metrics excluding the surface residual, model goodness-of-fit ranged from a mean R2 of 0.28 ± 0.13 (SD) at a daily temporal resolution, to 0.09 ± 0.06 for the 10-day temporal window (see ESM for table of model summaries). Models of surface residual generally had lower R2 values, with a mean of 0.02 ± 0.02. The sign of linear slope coefficients was consistent between the sexes and across foraging habitats at all temporal scales tested (Fig. 4). Coefficients were largest (i.e., relationships were steepest) at a daily temporal resolution, and decreased in strength with longer temporal windows (Fig. 4). In all cases, correlations between the dive behaviour metrics and foraging success were the inverse of those with seal drift rate (Fig. 3 and ESM Figs. S1–S4), and were of a lower magnitude.

Figure 3 Example model predictions (±95% confidence intervals) from the GAMMs fitted for the 1-day temporal window (see Fig. 4 for all windows).

Plots show left: the linear relationship between seal drift rate change (ΔDR) and each dive effort metric (descent rate, dive residual, ascent rate and surface residual); and right: the partial GAMM smooths for the nonlinear relationship between daily drift rate (DR) and each dive effort metric. Orange fits denote females, while green denotes males. Note that the sign of the linear slope coefficient is consistent across sex and habitat for all dive effort metrics except the surface residual.

Figure 4 Linear coefficient estimates with associated standard errors denoting the magnitude and sign of relationships between drift rate change and dive effort metrics.

For each dive effort metric (descent rate , dive residual, ascent rate), excepting surface residual, relationships are consistently positive or negative across temporal window, sex, and habitat. Coefficients with an associated “ns”, denote non-significant relationships. Two coefficients (descent rate and surface residual, for shelf-foraging females at the 10 and 2-day windows respectively) were marginally significant and are denoted by a “∼”. All remaining coefficients represented fitted relationships significant at the α = 0.05 level. In each panel, the dashed grey line denotes the location of the y-axis (zero line).

The sign of the above linear relationships between dive behaviour and foraging success depended on the dive metric in question. Mean descent and ascent rate both increased with increasing foraging success (significant positive association with ΔDR). Conversely, dive residual (an estimate of relative dive duration) decreased with increasing foraging success (significant negative association with ΔDR). The exception to these relationships was the final dive behaviour metric, surface residual (post-dive surface recovery time accounting for dive duration), which in most cases showed no significant association with foraging success. We note that while we do not report here on bottom phase metrics, we also tested hunting time (Heerah et al., 2015) and bottom time (Allegue et al., 2023) responses to foraging success and found them to be consistent with results obtained from the dive residual. To avoid overcomplicating the results and figures in the main text we have chosen to only report findings from the dive residual. Instead, we direct the reader to ESM section 1.2. for definitions of the hunting time and bottom time metrics, as well as model outputs and plotted results.

Discussion

Using a 20-year southern elephant seal tracking dataset, we tested for consistent signals in how air-breathing divers adjust their dive behaviour under changing foraging success. We quantified relationships between four dive behaviour metrics and an in situ metric of foraging success, and considered whether these relationships varied across sex and foraging habitat, after accounting for buoyancy effects. Overall, we found that buoyancy, our proxy of body condition, had the greatest magnitude of influence on dive behaviour. Indeed, as seal condition increased, so did their physiological capacity to perform longer dives with lower transit rates. Yet, after accounting for the controlling effects of buoyancy, we still found strong evidence of dive metric responses to foraging success. Notably, we found that the relationship between each of these dive metrics and foraging success were directionally consistent irrespective of sex or foraging habitat. Our results suggest a common pattern amongst sexes and habitats in which seals perform, on average, shorter, sharper dives during periods of high foraging success (Bestley et al., 2015; Thums et al., 2013). Below we contextualise these findings against the backdrop of prevailing foraging theory, and demonstrate that these relationships can be reconciled with predictions from the marginal value theorem.

Our results show that all seals increased transit rates (descent and ascent) in association with increasing foraging success. Surface residuals on the other hand did not decrease with increasing foraging success but instead remained invariable over time. Notably, we found no evidence of variable bottom phase responses to high foraging success across either sex or foraging habitat. Rather, all seals consistently decreased relative dive duration (dive residual) as foraging success increased, mirroring patterns observed for transit rates. Notably the direction of these relationships was consistent across a range of temporal scales, highlighting a general functional relationship in which dives become shorter and steeper with increasing foraging success. Explaining these results from an optimal foraging theory perspective, it is helpful to think of foraging success as an integrated measure of prey conditions encountered over multiple dives, akin to a neighbourhood of patches (MVT; Charnov, 1976). Under this model our reported relationships can be unified to represent how average dive metrics respond under changing neighbourhood quality, aligning with predictions from the MVT, wherein optimal stay time within a dive should decrease as neighbourhood quality increases.

Intuitively, we might expect that transit rates should increase with high foraging success, but this is rarely explicitly stated in optimal foraging theory aimed at diving predators (e.g., Thompson & Fedak, 2001). This prediction can however be inferred, assuming divers seek to minimise time between the surface and the dive bottom phase. Previous research has reported that transit rates are not linked to prey patch quality, but rather distance and buoyancy constraints (Gallon et al., 2007). We found that even after accounting for such constraints, transit rates still increased significantly with increasing foraging success. This builds support for similar work showing evidence that descent and ascent rates are strongly linked to prey encounter rates (Allegue et al., 2023; Bestley et al., 2013; Foo et al., 2016; Thums et al., 2013).

Increases in transit rates presumably aid seals in relocating prey more easily on subsequent dives (Foo et al., 2016), though given they occur concurrently with reductions in overall dive duration, they evidently do not serve to increase bottom time, as we might expect from the MVT. These findings seemingly counter the MVT expectation that an optimal forager should increase its dive bottom time when experiencing high energetic gains (Sparling et al., 2007; Thompson & Fedak, 2001). Importantly, evidence for this prediction deals with foraging success and dive behaviour at the scale of individual dives (Cornick & Horning, 2003) rather than across multiple dives, as we have done here. Following Charnov (1976), we would expect air-breathing divers to increase dive duration in those dives with high prey quality (e.g., through high encounter rates of many small, prey items, or through extended pursuits of large, mobile prey), but decrease average duration across a series of dives for which total foraging success is high. For instance, Watanabe, Ito & Takahashi (2014) found that Adélie penguins Pygoscelis adeliae increased dive bottom time as within-dive prey capture rates increased but decreased average bout-scale dive bottom time with increasing bout-level capture rates. Scaling up to a series of dives with high average foraging success, we can expect that the probability of prey encounters is high for any one dive. If a seal finds no or few prey at the bottom of a dive, it is more energetically efficient to give up early, because it will likely find more prey during the next dive (Thompson & Fedak, 2001). Similarly, if an animal sequentially encounters several high-quality patches its bottom time may decrease simply due to satiation and its stomach nearing or reaching capacity.

Physiologically, there is also an argument for shorter dives during periods of high foraging success. Because vertical transit rates increase with successful foraging, so too should oxygen consumption from faster swimming (although see also Boyd, Reid & Bevan, 1995). As there is a high probability of encountering prey in any given dive, seals probably end dives early (Viviant et al., 2016), so as not to exceed their aerobic dive limit, which would increase surface recovery time between dives. Alternately, when the probability of encountering prey is lower per dive, it may be advantageous for a seal to reduce transit rates (and associated metabolic costs), to free up oxygen capacity allowing for extended dive and search durations.

Unlike the other dive behaviour metrics, there was little change in surface recovery time with improving body condition and surface recovery remained largely uncoupled from foraging success. This may in part be explained by our estimated surface residual, which, while accounting for baseline recovery time associated with dive duration, may not fully account for variable energy expenditure associated with dives of the same duration but different depths (e.g., deep V-shaped dives vs shallow U-shaped dives). Elephant seals are generally considered exceptional for air-breathing divers with unique physiological adaptations for efficient oxygen recovery (Adachi et al., 2021; Hassrick et al., 2010). Recovery time is minimised following long, strenuous dives through enhanced oxygen loading efficiency, for instance through increase breathing rates (Hindell & Lea, 1998; Le Boeuf et al., 2000), which allows elephant seals to display very little variability in time spent at the surface. Our findings of invariable surface intervals, suggest that the seals tracked in this study dived well within their oxygen constraints and were not under any resource, oxygen/respiratory, stress. Alternatively, these seals may recover from an oxygen debt accumulated over a series of dives (e.g., Meir et al., 2009), through infrequent, long surface intervals not resolved in the mean surface residuals reported here. Nonetheless, the largely invariant surface recovery times in response to changing foraging success demonstrated in this study may not be true for smaller, shallower diving species with lower dive capacity. Indeed, in other species, animals are known to shorten surface recovery in good forage conditions to facilitate relocation of prey (Hanuise, Bost & Handrich, 2013). For these species, which must balance dive costs with surface recovery more actively, we would expect to find stronger links between surface recovery time and foraging success (e.g., Foo et al., 2016).

Overall, we found that the relationships between dive behaviour and foraging success were strongest at temporal windows of 1–2 days, matching similar work on this species (Allegue et al., 2023), and consistently weakened (i.e., estimated slopes flattened) as the temporal window increased. This may reflect changes in prey structuring as we move from meso- to basin scales. Southern elephant seals routinely travel between 30 and 100 km daily (Hindell et al., 2016), distances which are commensurate with the size of mesoscale oceanic structures (e.g., eddy diameters are 30–300 km; Patel et al., 2020). Mesoscale features are important for shaping prey distribution (Della Penna & Gaube, 2020; Green et al., 2020), and the density of prey is expected to be higher in or around eddies. It therefore makes sense that the strongest changes in behaviour evident in this study were for scales fine enough to resolve these patterns, and subsequent variability in foraging success. However, distances covered by seals increase as temporal windows get longer, effectively averaging out foraging success variability across different neighbourhoods of dives; this is why relationship strength for windows of 7–10 days approached zero. These results emphasise the importance of scale when representing ecological processes relevant to foraging animals (Levin, 1992).

While short-term increases in buoyancy can safely be interpreted as successful foraging (Arce et al., 2019; Biuw et al., 2007; Crocker, Boeuf & Costa, 1997), the opposite (loss of buoyancy equating to low foraging success) is not necessarily true. Decreases in buoyancy could also be related to recovery of lean tissue at the onset of the post-breeding/post-moult foraging trip, or with the late stages of foetus development in pregnant females (e.g., see Yong et al., 2024). This can lead to decreases in the relative proportion of lipid content, despite successful foraging. However, in the case of lean tissue recovery, recent work on northern elephant seals has shown that the majority of mass loss during fasting is fat, with skeletal muscle being largely preserved (Wright et al., 2020). Lean muscle takes proportionally longer to acquire which is why blubber is the primary mechanism for rapid mass gain. Given most of our observations occur several weeks and months after trip commencement, we expect muscle-related reductions in buoyancy to contribute only a small portion of our dataset. Similarly, in the case of pregnant females, embryo implantation only occurs around February (following a 3.5–4 month diapause; Atkinson, 1997), and we would expect significant gains in lean tissue associated with foetus development to only occur several months into the pregnancy (Yunker et al., 2005). As such, as with recovery of lean tissue, we suggest that reductions in buoyancy linked to foetus growth rather than poor energy gain represent a relatively small component of the greater dataset. Indeed, a post-hoc investigation using only positive ΔDR values (i.e., considering only cases where we were certain lipid content had increased) found the sign of our linear slope coefficients did not change, supporting the robustness of our reported relationships.

We used temporal windows as a representation of patch neighbourhoods, with changes in dive behaviour and foraging success reflecting movement between neighbourhoods. However, we have not explicitly considered movement processes to resolve the spatial scale of these neighbourhoods. Our findings suggest that dive behavioural responses to changes in foraging success are consistent and robust irrespective of whether spatial patterns are considered. Nonetheless, given the importance of travel time in foraging theory, the more explicit incorporation of spatial movements in future studies could provide more nuance to our understanding of these processes. For instance, following similar approaches as outlined in Thums et al. (2013) horizontal movements could be used to better define the scale and profitability of patch neighbourhoods in which seals forage. Similarly, without explicit incorporation of fine-scale spatial processes, we have used a fairly coarse classification of oceanic and shelf habitats. Future work could consider a more detailed analysis of individual dive characteristics (e.g., actively identifying benthic vs non-benthic dives; see McMahon et al., 2023) as well as fine-scale oceanographic features, to generate a more refined habitat characterisation. Such additions would allow further exploration of whether spatially resolved patch neighbourhood characteristics influence the relationships reported here.

We focused on the influence of bottom-up factors (prey conditions encountered) in shaping the optimal foraging behaviour of elephant seals. However, it is worth noting the importance of predation as an additional influencer on elephant seal dive behaviour. Elephant seals are regular prey of orcas (Orcinus orca; Jefferson, Stacey & Baird, 1991; Reisinger, e Bruyn & Bester, 2011), and it is known that they modify their diving in an effort to avoid being eaten. For instance, northern elephant seals defer sleep until they are below the photic zone and less vulnerable to visual predators (Kendall-Bar et al., 2023; LeBoeuf et al., 1986). Likewise, southern elephant seals when departing or returning to breeding and moulting sites tend to perform deep transit dives (>300 m), likely to avoid detection in areas where orcas are relatively abundant (Slip, Hindell & Burton, 1994). Therefore, prey distribution and abundance is clearly not the only factor at play for a seal attempting to forage optimally. Future work might seek to incorporate measures of predation risk within optimality models, to resolve how a seal’s dive behaviour would reflect its attempts to maximise energy gain whilst minimising predation risk.

The routine collection of position-time-depth information over the last few decades has led to the accumulation of a wealth of large and invaluable datasets on marine predator diving behaviour from remote parts of the world’s oceans (Carter et al., 2016). However, for satellite-relayed dive information, the necessary abstraction of dive profiles to a set of inflection points can impact interpretability (though information loss can be ameliorated using appropriate summarisation alogorithms; Photopoulou et al., 2015). High-resolution recorders such as accelerometers, jaw motion sensors and head-mounted cameras could be considered more effective for inferring fine-scale (individual dives–dive bouts) dive behaviours as they can compare dive shapes or metrics with direct estimates of feeding rates (e.g., Foo et al., 2016; Watanabe & Takahashi, 2013). Such investigations would also provide valuable insights into how dive behaviour responds to temporal changes in the type of prey encountered, something which we have not considered here. However, because of the high data overhead, such studies are generally restricted to short timescales, and species or annual periods for which the probability of recovery is high. Additionally, while these methods provide some indication of encounter rates, they do not necessarily imply foraging success (e.g., a prey capture attempt could involve a single small/large prey item or several prey).

Here, we demonstrate how in situ measures of foraging success can provide valuable interpretability to foraging behaviours inferred from heavily summarised dive profile data; particularly in tracking studies where there is limited capacity for device recovery. Our findings contribute greater understanding towards how air-breathing diving predators modify their behaviour in response to their foraging environment. Specifically, we show that dives become shorter and steeper with increasing foraging success, and that these patterns are ubiquitous, i.e., are not case specific or constrained to a particular sex or foraging habitat. Our results draw into question the traditional interpretation of optimal foraging theory that dive bottom time increases with higher prey encounters (Charnov, 1976; Thompson & Fedak, 2001), and is hence linked to higher foraging success. Generalising our observations from elephant seals could help towards inferences about foraging habitat in other diving animals that do not perform drift dives. Having said this, some caution may be warranted when applying our findings more broadly because the relationships we described show dive changes to foraging success, under the fundamental constraints of buoyancy. Future studies seeking further generalisation to estimate foraging success from dive behaviour metrics alone, could benefit by calculating the residuals of dive behaviour metrics around a fitted mean, itself representing buoyancy-controlled changes in dive capacity. Achieving this would provide scope for assessing individual-level habitat use, mapping variation in response to environmental covariates, and ultimately add capacity to current efforts assessing marine animal risk to climate and anthropogenic driven ecosystem change (e.g., Orgeret et al., 2022). Finally, future work could consider whether the nature of these dive behaviour–foraging success relationships shows evidence of variability over time. This would provide deeper insights into how seals might respond to changes in prey abundance and spatial structuring under a changing biophysical environment.

Supplemental Information

Supplemental Information 1 Supplementary Material

Additional Information and Declarations

Competing Interests

Author Contributions

Animal Ethics

Field Study Permissions

Data Availability

The authors declare there are no competing interests.

David B. Green conceived and designed the experiments, performed the experiments, analyzed the data, prepared figures and/or tables, authored or reviewed drafts of the article, and approved the final draft.

Sophie Bestley conceived and designed the experiments, performed the experiments, analyzed the data, authored or reviewed drafts of the article, and approved the final draft.

Clive R. McMahon conceived and designed the experiments, performed the experiments, analyzed the data, authored or reviewed drafts of the article, and approved the final draft.

Mary-Anne Lea conceived and designed the experiments, performed the experiments, authored or reviewed drafts of the article, and approved the final draft.

Robert G. Harcourt conceived and designed the experiments, performed the experiments, authored or reviewed drafts of the article, and approved the final draft.

Christophe Guinet conceived and designed the experiments, performed the experiments, authored or reviewed drafts of the article, and approved the final draft.

Mark A. Hindell conceived and designed the experiments, performed the experiments, analyzed the data, authored or reviewed drafts of the article, and approved the final draft.

The following information was supplied relating to ethical approvals (i.e., approving body and any reference numbers):

All tagging procedures were approved and executed under University of Tasmania Animal Ethics Committee guidelines, the Comité d’éthique Anses/ENVA/UPEC (no. APAFiS: 21375), or the Australian Antarctic Animal Ethics Committee (grant nos. AAS 2265 and AAS 2794).

The following information was supplied relating to field study approvals (i.e., approving body and any reference numbers):

Fieldwork was conducted under permit approval of, and supported by the respective national program administering each field site, namely the French Polar Institute Paul-Émile Victor (Iles Kerguelen; IPEV 109 & 1201), the Tasmanian Government (TFA 22459; Macquarie Island), the Australian Antarctic Division (AAS 2265 & AAS 2794; Davis Station and Casey Station) and the New Zealand Department of Conservation (SO-32768-MAR; Campbell Island).

The following information was supplied regarding data availability:

The data and code that support the findings of this study are available at Figshare: Green, David; Bestley, Sophie; McMahon, Clive; Lea, Mary-Anne; Harcourt, Robert; Guinet, Christophe; et al. (2024). Elephant seal dive behaviour responds consistently to changes in foraging success regardless of sex or ocean habitat. figshare. Dataset. https://doi.org/10.6084/m9.figshare.25029611.v4.

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
