# Peer review of "Elephant seal dive behaviour responds consistently to changes in foraging success regardless of sex or ocean habitat"

_PeerJ, doi:10.7717/peerj.20378_

## Round 0.1 · original submission · Minor Revisions

· Academic Editor

Minor Revisions

Dear authors

We have now received three in depth reviews of your work, which recognize the importance and contribution of your work, but suggesting some modifications before it can be accepted for publication. Please attend to the suggestions and reply to each argument raised by the reviewers. We are looking forward in receiving your reviewed article.

Reviewer 1 ·

Basic reporting

The authors use an impressive dataset to tackle a difficult but important question in the foraging ecology of air-breathing diving marine animals. I commend the authors for their creativity in calculating metrics to tease apart the drivers of dive behavior. The article is well written with a good background provided; the gap the authors are working to fill is clear.

I appreciated the figshare repository. I was able to run all the code and recreate the figures without issue. My one small comment regarding the scripts is that the figure numbering in the file names is incorrect (2-5 in the code, and 1-4 in the manuscript).

While I found the figure design clean and clear, I have several comments regarding the captions and labels. The PeerJ guidance is for the captions to be “self-contained”; to meet this standard, I ask that more detail be added to the captions. In the captions of Figures 2-4, the y-axis labels are described in the caption as “dive effort metrics.” To walk the reader through the figures, I suggest that all the y-axis variables be written out. Please also add units for the y-axis in both the figures and the captions.

I found the use of the term “variability” in the Figure 2 caption to be confusing. When I first read the caption, I expected the data points to represent a measure of variability. I would suggest being more straightforward in the caption writing and simply stating the relationship being displayed (e.g., change in mean daily drift rate across days since departure).

In Figure 3, I would like to see the meaning of the colors added as either a legend on the figure or written in the caption.

In Figure 4, I wonder if it would be possible to include a mention of “drift rate change” in the x-axis label? I also found the marginal significance symbol (“.”) hard to see on the plot; maybe the size could be increased or the symbol changed?

Experimental design

The metrics used are well described and cited, though please add units in the text. The model design seems appropriate.

L196-199: I would appreciate more justification for the dive filtering decisions described in this paragraph.

L236-238: On what scale were the locations classified? One per day? Or per dive?

L238-239: I think the addition of a table listing out all the models might help the reader keep track.

Validity of the findings

I have a significant concern regarding the framing and interpretation of the results. There is a significant mismatch in scale between the measure of “foraging success” (change in drift rate) and the research question. I struggle to see how a metric that averages net foraging success across at least one day connects to an individual’s perception of prey availability and decision making in the moment (which is the scale of foraging theory). The concept of a “neighborhood of dives” is mentioned several times. However, the scale of the “neighborhood” being assessed in this study is never described. From L91, I interpret “neighborhoods” to contain consecutive dives. But, from the methods, it appears that all dives within the temporal window are averaged regardless of their temporal distribution. Perhaps a time-series analysis approach incorporating both the timing and behavior of the dives would be more appropriate? Or some assignment of foraging bouts? Summaries of dive counts within the windows would be useful. Furthermore, although the authors test different temporal windows in the analysis, there is no mention of, or accounting for, spatial scale. In L359, the authors suggest that the changes in coefficient across time windows could reflect spatial scale, but spatial scale is not incorporated in the study. The incorporation of spatial movement would benefit the study, given the importance of travel time in foraging theory, which includes both vertical and horizontal movement in the marine environment. Spatial movement could also be useful in defining “neighborhoods”. Particularly if these seals can move 30-100km in a day (as mentioned in the discussion). Studying MVT for these marine predators is incredibly challenging, and I commend the authors' work to address these important theoretical questions using this dataset. I think MVT and the “neighborhood” concept could still be discussed if the spatiotemporal scale were better defined and incorporated into the analysis. Alternatively, I do think that a metric like change in drift rate can be used to address other important and interesting questions (e.g., how an individual changes its foraging behavior in response to internal state). This is a cool and interesting analysis, but the framing and research questions need to align better with the scale of the data and the analysis.

Regarding the interpretation of the model outputs. First, I would like to see a mention of model performance and fit in the main text. Second, I think the interpretation would be clarified by more discussion of the actual magnitude of these effects. Besides significance, the y-axis scales are small, and it’s hard to comprehend how big a 0.5 change in dive residual really is, for example.

I was a bit confused by the author's discussion of the surface residual result. I do not quite understand the justification in L343 regarding the population increase. Furthermore, I wonder if the lack of results could be affected by the averaging of the dives and the dive time filtering. Could the animals be accumulating oxygen debt over a series of dives with a long recovery (>10 minutes) afterwards? Also, while I realize these metrics cannot be measured from these tags, there are other fine-scale measures of physiology that could be changing instead of surface recovery time, such as inhalation duration or respiration rate.

Reviewer 2 ·

Basic reporting

Summary
This is a strong paper with important findings. My main suggestion for improvement concerns the flow of information. Several key background elements appear later in the Methods or Discussion, which I found left me with unresolved questions early on. Expanding the Introduction to include these concepts would strengthen the conceptual framework and make the paper easier to follow. In particular, the Introduction could more explicitly address: potential predator effects, seasonal variation in prey, and physiological consequences of body condition. I also provide line-by-line comments. I hope my comments are helpful, and I wish the authors well with their research.

General Comments
1. Mesopredator-top predator interactions: For mesopredators, fear may also modulate relationships with prey, as top predators can match prey distributions and increase pressure. Inconsistent relationships are often due to missing information on top-down or bottom-up pressures. For example, in the paragraph at lines 108-117, you note foraging costs and phenological variation, but what about predator avoidance? This could be an important consideration.
2. Physiological implications: How physiology changes with body condition is mentioned later, but introducing these nuances earlier would strengthen the framing.
3. Model validation: The Methods would benefit from a note on model checks, validation procedures, and diagnostics.
4. Seasonal prey shifts: If prey type changes seasonally, this could influence expected foraging behaviour; worth noting in the Introduction or Methods.
5. Habitat classification: The shelf vs. oceanic classification is quite coarse and does not account for fine-scale oceanographic features or prey field estimates. Consider acknowledging this limitation.
Line-by-line comments:

Line 32: This sentence could use some smoothing – maybe remove “in response” and the comma after successfully?
Line 45: clarify “elephant seal” – I think the generality in the next sentence is ok, but would keep it species-specific here
Line 171: Did you check the one-step-ahead residuals (aniMotum::osar()) for goodness of fit?
Lines 174-180: consider moving to intro (just a suggestion – it's lots of background info for the methods, but I think that’s just personal preference)
Lines 197-199: any rationale and references for these numbers? Reason (1) is self-explanatory, but a little explanation and reasoning for reasons 2 through 5 would be helpful.
Lines 216: Why running means and not just plain lags? Some rationale and references for the running mean would be helpful.
Line 238: Should “we fitted separate models…” be the topic sentence for the next paragraph?
Line 241: DR is defined above
Lines 224-234: I think moving (and refraining to be more for background) this to the intro would help with a lot of the curiosities about nuances that I had when reading the intro
Line 235: again feels like intro material
Line 237: reference/reasoning for 1500 m cutoff?
Line 255: nice
Line 279-281: I think that would be a nice addition, unless there’s a reason not to report them here?
Line 292: Could you add a little to how these results are important for our theoretical understanding of dive-success relationships, instead of the sentence on “we make sense of these findings” sentence
Lines 301-306: I think an important caveat is what happens when the near and reach stomach capacity?
Line 305: quality or density/quantity?
Line 329: another spot to mention nearing stomach capacity
Lines 316-342: missing a ton of references for theory and empirical/critical work that’s been done on the theory
Line 387: reference to an appendix table?
Line 407: ref for traditional view?
Fig 1: This is a nice colour palette, but I find the orange on grey pretty hard to see in spots where there’s lower track density
Fig 2A: What’s with the horizontal gap around drift rate 0 to -0.1? Is that an artefact of drift rate calculation?
Fig 2: Are these panels cropped? The data seems to go right to the edge, and I wonder if some data is cut out?
Fig S3: qqplots look kind of weird – did you try to get better fits?

Experimental design

-

Validity of the findings

-

·

Basic reporting

The manuscript by Green et al. is extremely well written, clear, and has an excellent structure. The length is completely appropriate, and I commend the authors on writing an interesting and readable study that captures a unique dataset on Southern elephant seals (>600 individual tag efforts, >20 years) and tests clear hypotheses about dive effort and foraging success in the context of theory. The figures are very clear and meet the expectations for quality among the community. There is much for many readers to like, so I think this manuscript should be readily cited.

Experimental design

Terrific, detailed, clear, and self-explanatory. Green et al. provide full context and details about methodology, experimental approach, and relevant permits and authorizations. The relevant sections are all appropriately cited.

Validity of the findings

This paper does an excellent job of framing a phenomenon -- diving in breath-holding animals to forage -- and provides a test using an interesting dataset (decadal in scale, hundreds of replicates) in the context of theory. Green et al. make it clear that there are competing theoretical frameworks (i.e., optimal foraging theory and marginal value theorem) to explain the relative changes in dive behavior for foraging success, and use the depth and quality of their dataset to test various predictions, including differences attributable to sexual dimorphism (something important when considering elephant seals). Overall, their conclusions crisply track with support from their analyses.

Additional comments

I really enjoyed reading this paper, as an outsider to behavioral ecology and just slightly adjacent to the world of diving physiology. For example, their finding that Southern eseals reach neutral buoyancy around their third month at sea is a fabulous nugget of insight, and I suspect there are other particular findings that the in-group of the marine mammal diving community will find readily useful and citable. This paper should definitely be published in PeerJ.

I think an insider might consider this paper perfect (i.e., acceptable as is), but I think there are some very minor points to consider that would be easy fixes should the authors agree.

First, there are other costs to foraging that aren't entirely tied to prey patchiness and abundance. In the North Pacific, predation risk (by killer whales) appears to be a strong driver for diving behavior in congener elephant seals, but it's really not mentioned in this manuscript. I think the Discussion would benefit from a brief mention of this obvious risk, inter alia, and then offer a bit more for why or why not this risk might be present for seals below the Circum-Antarctic convergence. After all, killer whales (types A and Bs) are abundantly present both in time and space of the dataset under consideration.

Second, the sexual dimorphism in elephant seals is not really emphasized in the Introduction. It's one of the great features of the dataset that Green et al. analyze (e.g., l. 297), but there's not really a logic to why it should be relevant in the study. I suggest a few words (or a sentence) emphasizing this point to improve the rationale for non-marine mammal specialists near ll. 118-126.

Lastly, the authors readily note that the spatial scale of mesoscale oceanic structures matches the scale of Southern eseal foraging (ll. 360-369), but there's no obvious consideration of temporal oceanographic changes in the Southern Ocean over the course of their decadal scale dataset that might have affected prey patchiness and abundance. I think there should be at least a few sentences spotlighting why these physical factors might or might not have had an effect on the big picture of eseal foraging, perhaps if necessary as its own paragraph. In other words, space and time factors both deserve full consideration here.

Minor items:
ll. 59-60, use a colon instead of a semi-colon, and then separate items by semi-colon. Edit as "...three distinct phases: a bottom phase associated with active foraging; ...[repeat]"

ll. 74 and 79, words wasted and saved should probably not be in double quotes, as that implies text citation, suggest single quotes.

Overall, this manuscript is essentially ready for publication with the consideration of these minor comments; I do not need re-review, and congratulate the authors on a job well done.

---

## Round 0.2 · accepted · Accept

· Academic Editor

Accept

Dear authors

Thanks for addressing all reviewer's comments with clarity and I congratulate you on this important work.

Best regards